# Tracking down the White Plague: The skeletal evidence of tuberculous meningitis in the Robert J. Terry Anatomical Skeletal Collection

Olga Spekker[1]*, David R. Hunt[2], László Paja[1], Erika Molnár[1]ᵒ, György Pálfi[1]ᵒ, Michael Schultz[3]ᵒ

**1** Department of Biological Anthropology, University of Szeged, Szeged, Hungary, **2** Department of Anthropology, National Museum of Natural History, Smithsonian Institution, Washington, District of Columbia, United States of America, **3** Institut für Anatomie und Embryologie, Zentrum Anatomie, Universitätsmedizin Göttingen, Göttingen, Germany

ᵒ These authors contributed equally to this work.
* olga.spekker@gmail.com

**Data Availability Statement:** All relevant data are within the paper and its Supporting Information files.

## Abstract

Paleopathological diagnosis of tuberculosis (TB) essentially relies on the identification of macroscopic lesions in the skeleton that can be related to different manifestations of TB. Among these alterations, granular impressions (GIs) on the inner skull surface have been considered as pathognomonic features of tuberculous meningitis (TBM). GIs may be established by pressure atrophy of the tubercles formed on the outermost meningeal layer during later stages of TBM. Although GIs were used as diagnostic criteria for TBM in the paleopathological practice since the late 20$^{th}$ century, their diagnostic value has been questioned. To contribute to strengthening the diagnostic value of GIs, a macroscopic investigation–focusing on the macromorphological characteristics and frequency of GIs–was performed on skeletons of known cause of death from the Terry Collection. The $\chi^2$ analysis of our data revealed that GIs were significantly more common in individuals who died of TB than in individuals who died of non-TB causes. Furthermore, GIs were localized on the inner surface of the skull base and of the lower lateral skull vault. The localization pattern and distribution of GIs on the endocranial surface resemble that of the tubercles observed in the affected meninges during the pathogenesis of TBM. Our results strengthen the tuberculous origin of GIs and imply that they can be considered as specific signs of TBM. Therefore, GIs can be used as diagnostic criteria for TBM in the paleopathological practice, and the diagnosis of TBM can be established with a high certainty when GIs are present in ancient human bone remains.

## Introduction

Tuberculosis (TB), commonly referred as the "White Plague", is one of the oldest known infectious diseases that has been afflicting humans and animals for thousands of years [1–2]. It is caused by several pathogenic mycobacterial species that belong to the *Mycobacterium*

**Funding:** This work was funded by the Hungarian State Eötvös Fellowship 2016 (77466) of the Tempus Public Foundation, the NTP-NFTÖ-16 (1116) of the Hungarian Ministry of Human Capacities & Human Capacities Grant Management Office, and the University of Szeged Open Access Fund (4393) to OS. The National Research, Development and Innovation Office (Hungary) (K 125561) provided funding for GP. The funders had no role in study design, data collection and analysis, decision to publish, or preparation of the manuscript.

**Competing interests:** The authors have declared that no competing interests exist.

*tuberculosis* complex (MTBC), with *M. tuberculosis (sensu stricto)* being the most common cause of TB in humans [3–5]. TB bacteria are usually transmitted by the airborne route; therefore, the disease primarily affects the lungs (i.e., pulmonary TB) [5–9]. Nonetheless, the hematogenous or lymphogenous spread of the pathogens to other parts of the body, including the skeleton or the central nervous system (CNS), results in the development of extra-pulmonary TB [6,10]. Different manifestations of extra-pulmonary TB (e.g., miliary TB, skeletal TB, and CNS TB) together constitute up to 25% of all active TB cases [10]. TB bacteria that reach the alveoli in the lungs may be eliminated by the host's immune system [5–6,9]. However, in most people, the pathogens are able to escape eradication and invade into the lung parenchyma [5–8,11]. The presence of TB bacteria triggers the recruitment of an increasing number of immune cells (e.g., macrophages and lymphocytes) to the sites of infection [5–8]. Ultimately, there is the formation of tuberculous granulomas–also known as tubercles–that are the hallmark features of TB [5–9,11]. Tubercles provide an isolated microenvironment in which host cells interact to control and prevent dissemination of the infection [5–8]. However, tubercles also function as a survival niche in which TB bacteria can replicate or persist in a dormant state within the lung tissue until opportunity arises for them to reactivate and spread [5–8]. In the minority of people affected (∼10%–mainly in infants and children), tubercles fail to contain the infection and TB bacteria can disseminate throughout the lung or into other parts of the body [5–8,11]. The infection progresses into active TB disease (i.e., the infected person becomes symptomatic and contagious), usually within the first two years after exposure [5–8,11]. In approximately 90% of the cases, TB infection is latent (i.e., the infected person is asymptomatic and not contagious) [5–9,11]. In latent TB infection (LTBI), pathogens remain dormant within the tubercles for a long time (even for a lifetime), with subsequent reactivation occurring in about 5–15% of people with LTBI [5–7,9]. The processes underlying the reactivation of TB disease from latency are still poorly understood, and often, no known or suspected risk factors can be identified [5–9,11]. Nonetheless, certain factors, including malnutrition, human immunodeficiency virus (HIV) infection, and *diabetes mellitus*, have been mentioned as potential risk factors for LTBI reactivation in the modern medical literature [8,11].

The growing HIV/acquired immune deficiency syndrome (AIDS) pandemic, as well as the emergence of multidrug-resistant TB have played a significant role in the resurgence of TB since the late 1980s [4,9–10]. Subsequently, the disease has been declared a global public health threat by the World Health Organization (WHO) in 1993 [12]. According to the WHO estimates, approximately 1.7 billion people– 23% of the total population of the world–have LTBI today [13]. In 2017, there were approximately 10.0 million incident cases of active TB globally and the disease remained one of the top ten causes of death and the leading cause of death from a single infectious agent (ranking above HIV/AIDS), with accounting for 1.6 million deaths [13]. The global public health emergency presented by TB today has sparked a renewed interest and funding to the research of the disease and of its etiological agents, including science projects concerning the origin and evolutionary history of the MTBC, as well as the paleopathological diagnostics for TB [5,13–14].

The paleopathological research of TB is essentially based upon the macromorphological diagnosis of the disease in ancient human bone remains [14–15]. It provides invaluable data on the different manifestations of TB, as well as on the effects of the disease upon human mortality and morbidity around the world throughout prehistoric and historic times [14–15]. Using modern medical knowledge, paleopathologists endeavor to establish a retrospective diagnosis of TB by macroscopically identifying pathological conditions (e.g., spinal TB and TB arthritis of the large, weight-bearing joints) in human skeletons that may be related to the disease [16–17]. However, utilization of modern diagnostic criteria for TB in the paleopathological practice may not always be appropriate. On the one hand, probable TB-related bony

changes observed in recent cases may differ from those detectable in ancient human bone remains (due in part to the introduction of antibiotics in the treatment of the disease) [16,18–20]. On the other hand, in living TB patients, bony changes cannot be surveyed with macro-morphological methods but with medical imaging techniques, such as X-ray radiography, computed tomography (CT), and magnetic resonance imaging (MRI), only [18–19]. Neverthe-less, subtle bony alterations are mostly impossible to be visualized by modern imaging meth-ods [18–19]. Therefore, subtle bony changes are not relevant to the diagnosis of TB in living patients and are not described as diagnostic criteria for the disease by physicians in the modern medical literature, even if they can be potentially important elements of TB identification for paleopathologists [16,19,21].

The assessment of TB frequency in past human populations has traditionally relied upon the paleopathological diagnosis of spinal TB and/or TB arthritis of the large, weight-bearing joints only [16,18–22]. In modern clinical assessments, osteoarticular TB occurs in less than 2% of all patients with active TB. In consideration that skeletal TB is identified in about 3–5% of all the TB cases in prehistoric and historic times, it is difficult to assess the true frequency of the disease in human osteoarchaeological material from the pre-antibiotic era based only on the above-mentioned diagnostic criteria [17–18,20,22].

Since the late 20th century, a number of studies [e.g., 15–32] were performed on osteoarch-aeological series and documented skeletal collections to contribute to the establishment of a more reliable and accurate paleopathological diagnosis of TB and to the assessment of a more relevant disease frequency in past human populations. These studies [e.g., 15–32] have revealed a positive correlation between different manifestations of TB (e.g., skeletal TB, pulmo-nary TB and/or TB pleurisy, and TB meningitis (TBM)) and subtle bony alterations. Schultz [24–26] and Schultz & Schmidt-Schultz [27] have identified granular impressions (GIs) on the inner surface of the skull as one of these bony changes. GIs are small (0.5–1.0 mm in diameter), relatively shallow (less than 0.5 mm in depth), roundish impressions with smooth margins and walls [24–28,30]. They represent a pathological process that affects only the superficial part of the inner lamina of the skull with no diploic and/or ectocranial involvement [24–28,30]. GIs generally appear as isolated or confluent lesions grouped in clusters on the inner surface of the skull base or sometimes of the lower lateral skull vault: they are particularly situated in the orbital part of the frontal bone, the greater wings of the sphenoid bone, the squamous part of the temporal bones, and the lateral and squamous parts of the occipital bone [24–28]. Accord-ing to the results of Templin [28], Templin & Schultz [29], Schultz [24–26], and Schultz & Schmidt-Schultz [27], GIs may be established by pressure atrophy of the tubercles formed in the *dura mater* during later stages of TBM [24–29]. By using light and scanning electron microscopy, the *intra vitam* character of GIs and the mechanism of their origin can easily and convincingly be described in macerated bone specimens [24–29]. GIs were named as "sharply demarcated erosive defects" by Hershkovitz and his colleagues [31], but the lesions have an erosive macroscopic appearance with more irregular shape and sharper walls and margins only when the pathological process progresses and the tubercles resulting in the impressions become caseous. (It should be noted that although having very similar names in the scientific literature, granular impressions and granular foveolae (i.e., impressions of the arachnoid gran-ulations) should not be mistaken for each other, since they refer to two different lesion types affecting the inner surface of the skull.) GIs were described by Schultz [24–26] and Schultz & Schmidt-Schultz [27] as pathognomonic features of TBM; however, their diagnostic value has been questioned [33].

In the first half of the 20th century, bony changes associated with TBM were distinctly described in the pathological literature [e.g., 34–36]. These literature data can help paleopa-thologists in the establishment of a more reliable diagnosis of TBM and in the assessment of a

more relevant frequency of TBM in human osteoarchaeological material. Besides the pathological literature from the first half of the 20th century, the detailed analysis of well-documented collections of pre-antibiotic era skeletons (e.g., Hamann–Todd Human Osteological Collection, Robert J. Terry Anatomical Skeletal Collection, and Coimbra Identified Skeletal Collection) serves as a unique and important basis for determining the appropriate paleopathological diagnostic criteria for TB in past human populations, since 1) bone remains of individuals identified to have died of TB and not treated with antibiotics may exhibit similar TB-related bony changes to those of observable in skeletons of people who lived in the past; 2) in contrast to living TB patients, skeletons of known cause of death can be surveyed not only with medical imaging techniques but also directly with macromorphological methods; and 3) even subtle bony changes can be recognized in them [16–17,19,21,37]. In the last three decades, the Terry Collection has been used to define and refine paleopathological diagnostic criteria for TB in several studies [e.g., 18,22,38–40]; nonetheless, GIs were beyond the scope of the aforementioned research projects.

The main aim of our study is to expand knowledge and understanding about the development of GIs and to improve their paleopathological interpretation, along with strengthening their diagnostic value in the identification of TBM in human osteoarchaeological material. This is accomplished by presenting results of a macroscopic investigation–focusing on the macromorphological characteristics and frequency of GIs in skeletons of known cause of death from the Terry Collection. The macroscopic examination was performed on all individuals recorded to have died of different manifestations of TB in the Terry Collection. The control group consists of randomly selected individuals from the Terry Collection, identified to have died of causes other than TB.

The objectives of our paper are:

1. To macroscopically evaluate the selected skeletons from the Terry Collection for the presence of GIs;

2. To compare the frequencies of GIs between individuals recorded to have died of TB versus those identified to have died of causes other than TB;

3. To macromorphologically characterize GIs regarding the localization, extent, and number of lesions on the affected cranial bone(s); and

4. To evaluate the diagnostic value of GIs.

## Materials and methods

### Materials

The Robert J. Terry Anatomical Skeletal Collection–currently curated in the Department of Anthropology at the National Museum of Natural History (Smithsonian Institution, Washington, DC, USA)–consists of 1,728 human skeletons, mostly from the pre-antibiotic era [37]. For each individual, there is a series of documentary forms providing various biographical information (e.g., age at death, sex, and cause of death) and basic anthropological data [37]. The Terry Collection serves as an invaluable resource for anthropological and medical research, including defining and refining diagnostic criteria for specific infectious diseases, such as TB, in osteoarchaeological series from the pre-antibiotic era [37].

As part of a comprehensive research project [41], macromorphological characteristics, frequencies, and co-occurrences of pathological alterations probably related to different manifestations of TB were evaluated on all individuals (N = 302) recorded to have died of TB (e.g.,

pulmonary TB, miliary TB, peritoneal TB, and skeletal TB), and on a control group consisting of randomly selected individuals (N = 302) from the Terry Collection, identified to have died of non-TB (NTB) causes (e.g., other infectious diseases, cardiovascular problems, cancer, and external causes, such as suicide or homicide). From the 604 skeletons, 177 were excluded from the examination considering GIs: the skullcap was missing in two cases, the skull was not sectioned in a further 173 cases, and age at death was uncertain in two additional cases. The remaining sample consisted of 427 skeletons. The seven old adolescent (16–19 years old; three males and four females) and 420 adult (≥20 years old; 272 males and 148 females) individuals with skulls sectioned in the transverse plane (and occasionally also in the mid-sagittal plane) were divided into two main groups on the basis of their causes of death:

- TB group, consisting of 234 individuals (169 males and 65 females) identified to have died of TB, with age at death ranging from 16 to 81 years (S1 Table and S1A Fig); and

- Control (NTB) group, composed of 193 individuals (106 males and 87 females) recorded to have died of causes other than TB, with age at death ranging from 20 to 90 years (S2 Table and S1B Fig).

## Methods

No permits were required for the described study, which complied with all relevant regulations. The endocranial surface of the 427 selected skulls was macroscopically surveyed for the presence of GIs. To reduce the risk of being biased, the study personel had no information on the cause of death of the examined individuals during the macromorphological evaluation of the 427 selected skulls. A lamp was always positioned at a distance of a few centimeters from the bone surface, since the examined bony changes can have a very subtle appearance that makes their detection difficult. For each selected individual, detailed written and pictorial descriptions of all observed GIs were made on a data collection sheet prepared for the current research project. The affected cranial bone(s) (considering the left and right greater wings of the sphenoid bone as two separate bones); the number of detected lesions in the affected cranial bone(s) (unifocal or multifocal); and the extent of the endocranial surface area the observed lesion(s) covered (x) in the affected cranial bone(s) (4-level scale: 1) $x < 25\%$, 2) $25\% \leq x < 50\%$, 3) $50\% \leq x < 75\%$, and 4) $75\% \leq x$) were also recorded.

After the detailed macromorphological evaluation of the 427 selected skeletons, all collected information was entered into a spreadsheet in Microsoft Office Excel 2016, and subsequent statistical analysis of the data was performed: absolute and percentage frequencies of GIs were calculated in both the TB group and NTB group; and $\chi^2$ testing of the data to determine the significance of difference (if any) in frequencies of GIs between the two groups was undertaken using the MedCalc statistical software package.

## Results

During the macroscopic investigation, GIs were detected in 17.33% (74/427) of the skeletons examined–in 29.06% (68/234) of the TB group and in 3.11% (6/193) of the NTB group. The $\chi^2$ testing of the frequencies of GIs in individuals with TB as the cause of death and individuals with NTB causes of death revealed a statistically extremely significant difference between the two groups ($\chi^2$ = 47.922, df = 1, P<0.0001).

From a total of 68 individuals with GIs in the TB group, 53 died of pulmonary TB (S1 Table). Five additional individuals died of other types of TB, such as skeletal TB (three cases), TBM (one case), and peritoneal TB (one case); whereas in the remaining ten cases, the type of

TB as the cause of death was not specified on the morgue record and/or death certificate (S1 Table). Among the NTB causes of death of individuals with GIs, cardiovascular problems (three cases), cancer (two cases), and peritonitis (one case) were recorded (S2 Table).

Concerning the localization of GIs, the most commonly affected area of the inner surface of the skull was the squamous part of the occipital bone (Figs 1 and 2A) in both the TB group (62/68, 91.18%) (S3A Table) and NTB group (6/6, 100.00%) (S3B Table). Furthermore, GIs were quite often observed in the orbital part of the frontal bone (TB group: 32/68, 47.06%; NTB group: 4/6, 66.67%) (Figs 1 and 2B and S3A and S3B Table) and in the squamous part of the left (TB group: 20/68, 29.41%; NTB group: 2/6, 33.33%) and right (TB group: 20/68, 29.41%; NTB group: 3/6, 50.00%) temporal bones (Figs 1 and 2C and S3A and S3B Table). Occasionally, the involvement of the left (TB group: 10/68, 14.71%; NTB group: 1/6, 16.67%) and right (TB group: 19/68, 27.94%; NTB group: 1/6, 16.67%) greater wings of the sphenoid bone (Figs 1 and 2D and S3A and S3B Table), as well as the left (TB group: 6/68, 8.82%; NTB group: 1/6, 16.67%) and right (TB group: 10/68, 14.71%; NTB group: 1/6, 16.67%) parietal bones (predominantly along the squamous suture) (Figs 1 and 2C and S3A and S3B Table), was also registered. In both groups, less than four cranial bones (considering the left and right greater wings of the sphenoid bone as two separate bones) were simultaneously affected by GIs in approximately two-thirds of individuals (TB group: 49/68, 72.06%; NTB group: 4/6, 66.67%).

Regarding the number of presented lesions among individuals identified to have died of TB, GIs were particularly recorded as multifocal bony changes in the occipital (53/62, 85.48%) and frontal (24/32, 75.00%) bones, as unifocal alterations on the left (9/10, 90.00%) and right (14/19, 73.68%) greater wings of the sphenoid bone and in the left (4/6, 66.67%) and right (6/10, 60.00%) parietal bones; whereas the frequencies of unifocal (9/20, 45.00%) and multifocal (11/20, 55.00%) GIs were similar in both temporal bones (S3A Table). Among individuals recorded to have died of causes other than TB, only two GIs involving the left and right greater wings of the sphenoid bone were registered as unifocal alterations (S3B Table). As for the extent of the detected lesions, the majority of GIs observed in the TB group covered less than one-fourth of the endocranial surfaces in all cranial bones examined (S3A Table). Nonetheless, the extent of GIs in the squamous part of the right temporal bone and on the right greater wing of the sphenoid bone exceeded one-fourth of the inner surfaces quite often: in 30.00% (6/20) and 42.11% (8/19) of cases, respectively (S3A Table). In the NTB group, only three GIs detected on the left and right greater wings of the sphenoid bone and in the squamous part of the occipital bone covered more than one-fourth of the endocranial surfaces (S3B Table).

## Discussion and conclusions

In the paleopathological literature, GIs (sharply demarcated erosive defects [31]) on the inner surface of the skull have been considered as pathognomonic features of TBM–established by pressure atrophy of the tubercles formed in the *dura mater* (in close vicinity to blood vessels affected by TBM) during later stages of the disease [24–27]. However, the diagnostic value of GIs in the paleopathological identification of TB has been questioned: Roberts and her colleagues [33] have argued against that GIs can be of tuberculous origin. Their basis of argument is that: 1) TBM is not always accompanied by the formation of meningeal tubercles that could result in pressure atrophy on the inner surface of the skull, and consequently, the development of impressions; 2) in cases with macroscopically visible meningeal tubercles, no endocranial changes were described in the modern medical literature; 3) the typical course of TBM is not long enough to allow bony changes to occur on the inner surface of the skull; and 4) only a weak association between the presence of GIs and TB have been found by Hershkovitz and his

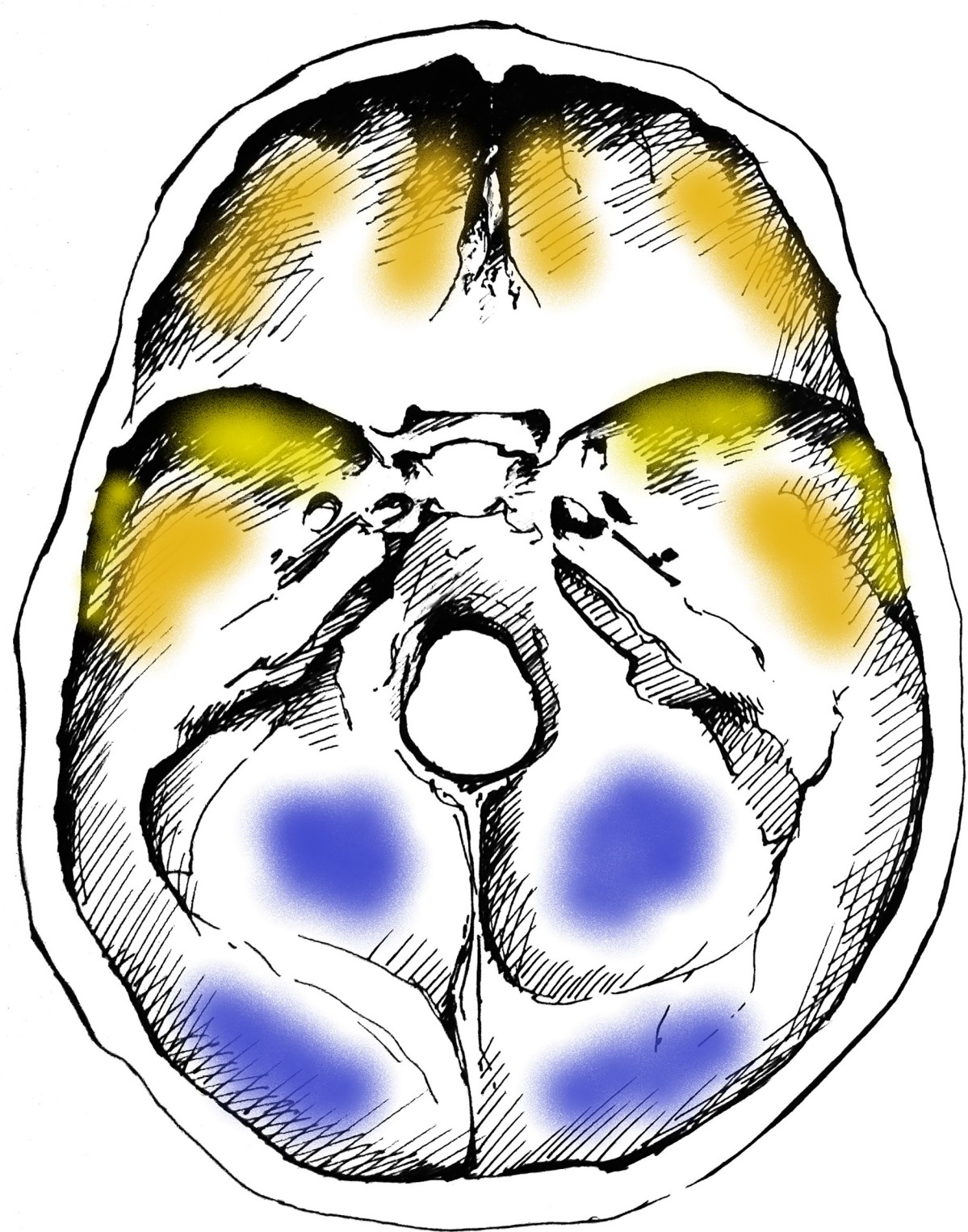

**Fig 1. Typical localizations of GIs on the inner surface of the skull base.** Blue: most commonly affected areas, orange: commonly affected areas, and yellow: less commonly affected areas (drawing by Luca Kis).

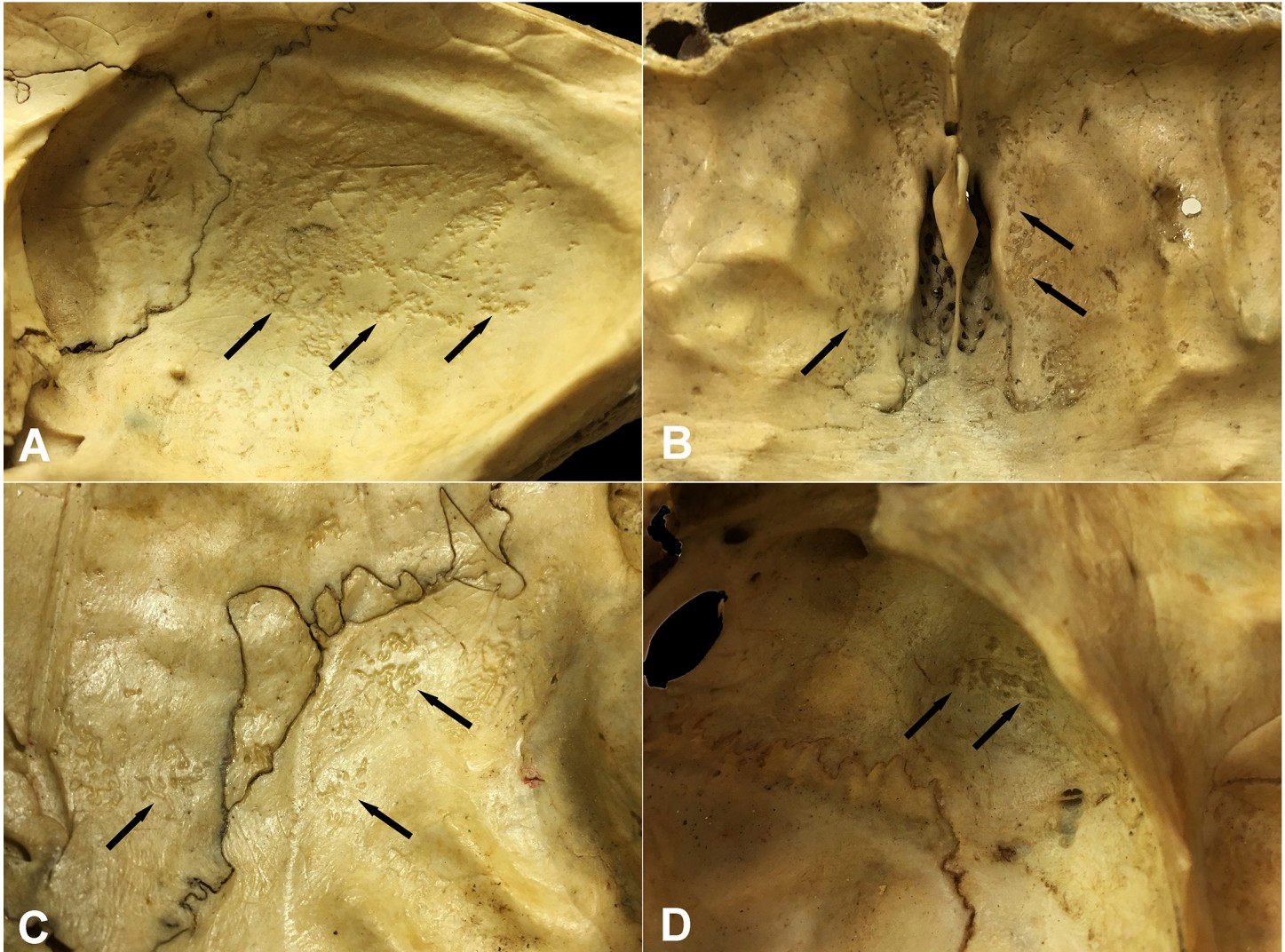

**Fig 2. Typical localizations of GIs on the inner surface of the skull.** A) The squamous part of the occipital bone (Terry No. 562, 17-year-old, female, died of pulmonary TB), B) The orbital part of the frontal bone (Terry No. 933R, 40-year-old, male, died of peritoneal TB), C) The squamous part of the temporal bone and the area along the squamous suture on the parietal bone (Terry No. 522, 30-year-old, male, died of pulmonary TB), and D) The greater wing of the sphenoid bone (Terry No. 566, 40-year-old, male, probably died of TB).

colleagues [31] during the examination of skeletons of known cause of death from the Hamann–Todd Collection. (However, their investigations focused not on GIs but on *serpens endocrania symmetrica*, another endocranial alteration type that may be associated with TBM but not pathognomonic to the disease.) According to Roberts and her colleagues [33], further investigations on GIs in skeletons of known cause of death or in skeletons from osteoarchaeological series with an independent confirmation of the diagnosis of TB (e.g., by light and scanning electron microscopy [27] or by biomolecular methods, such as ancient DNA, lipid biomarker, and extracellular matrix (ECM) protein [42] analyses) are needed to clarify the exact etiology of GIs.

Besides paleopathological evaluation of skeletons, detailed knowledge of the pathogenesis of CNS TB forms, especially of TBM, is essential to gain a better understanding of the pathological processes that underlie the development of GIs. CNS TB (one of the most devastating clinical manifestations of TB) occurs in approximately 1% of all active TB cases and accounts

for 5–15% of extra-pulmonary TB cases [43–48]. CNS TB usually results from hematogenous dissemination of TB bacteria from a primary location outside the CNS (such as the lungs or the gastrointestinal tract) and is characterized by a slowly progressive granulomatous inflammatory reaction that may affect the meninges, or the brain or spinal cord parenchyma [43,49–50]. The disease develops in two stages [43,45,48,51]. The initial stage involves the formation of small (0.5–2 mm) tubercles – also known as Rich foci – around TB bacteria deposited in the CNS via blood circulation during or shortly after the bacteremic stage of primary infection or late reactivation of TB elsewhere in the body [43–46,48,51–54]. Following their establishment, Rich foci may remain dormant for many years [43–45,52–53]. Later, the enlargement or rupture of one or more Rich foci results in the development of different types of CNS TB (e.g., TBM, tuberculomas, and TB abscesses) [43,46,48–49,52]. The most common form of the disease is TBM that accounts for 70–80% of all cases with CNS TB [44,49]. TBM usually develops subsequent to the rupture of one or more meningeal, subpial, and/or subependymal caseating Rich foci into the subarachnoid space or into the ventricular system, both occupied by the cerebrospinal fluid (CSF) [43–45,49,52–56]. The release of sufficient numbers of TB bacteria into the CSF triggers the onset of diffuse granulomatous inflammation of the leptomeninges (i.e., the *pia* and *arachnoid mater*), with a strong predilection for the basal areas of the brain [43–45,49,52–55]. In earlier stages of the disease, the pathological process extends predominantly along the blood vessels of the leptomeninges (particularly of the *pia mater*) [34]. TBM is characterized by the formation of tubercles on the *pia* and *arachnoid mater* [34,57]. Nonetheless, not only the leptomeninges but additionally, the outermost meningeal layer (i.e., the *dura mater*) that is directly adherent to the inner surface of the skull, can be affected by tubercles in later stages of TBM [35,57]. Besides the small tubercles formed in the meninges, characteristic pathological features of TBM include enhancing basal meningeal exudate, progressive hydrocephalus, and vasculitis of blood vessels adjacent to or traversing the exudate [34,43,45,52,55–56]; as a rule, the formation of tubercles precedes the development of the latter ones [35].

One of the most important risk factors for TBM is age [45–47]. In low-income and middle-income countries with a high incidence of TB, children under the age of five years represent the most vulnerable group affected by the disease, usually developing TBM within 3–6 months of primary infection [45,52–53,58–60]. However, in high-income countries with a low incidence of TB, TBM occurs predominantly in adults, particularly in immigrants from TB-endemic regions of the world and in HIV-positive people, who are five times more likely to develop the disease than HIV-negative individuals [43,45–46,53,58–60]. In adults, TBM usually results from the reactivation of dormant Rich foci, often many years after the primary infection [58]. At any age, TBM is one of the most severe extra-pulmonary manifestations of TB, with high short-term mortality and substantial excess morbidity among survivors: approximately one-third of the affected individuals die of the disease and up to one-half of the survivors remain with serious neurological sequels, despite the initiation of anti-tuberculosis therapy [43–44,48,58,61]. Early, accurate diagnosis and prompt, adequate treatment are crucial in determining the clinical outcome of TBM [43, 53,56,59,62]. If left untreated, TBM usually leads to death within 4–6 weeks after the onset of its symptoms [63]; nonetheless, in some cases [e.g., 64–70], the disease has a protracted course that can last for several months or even years. Therefore, in such cases, the duration of the disease is long enough to allow bony changes to occur on the inner surface of the skull.

Although no endocranial alterations were described in cases with macroscopically visible meningeal tubercles in the current medical literature, it does not mean that GIs cannot be considered as diagnostic criteria in the paleopathological practice. On the one hand, bony changes associated with TBM were distinctly described in the pathological literature from the first half of the 20th century (preceding the introduction of antibiotics in the management of TB) [e.g.,

34–36]. At autopsy of TBM patients, groups of isolated but mostly confluent, small, and dimpled impressions established by pressure atrophy of the tubercles (i.e., GIs), vestiges of hemorrhages developed in close vicinity to the affected blood vessels (i.e., abnormal blood vessel impressions and periosteal appositions), and certain roughnesses indicating characteristic resorption of the bone tissue (i.e., very flat and small erosive bone loss that can be recognized only with low-power microscopy) were observed on the endocranial surface of the skull base and of the lower lateral skull vault after the removal of the *dura mater* [34–36]. However, autopsy practices have changed over time, and in the present, the *dura mater* is not completely removed from the basal areas of the skull; thus, there is no detection of the aforementioned bony changes on the endocranial surface. Additionally, the identification of TBM in living patients is usually based on clinical signs and symptoms, CSF findings, and radiological characteristics [43–47,52–53,55]. Similar to periosteal new bone formations on the visceral surface of ribs [18–19], GIs have a very subtle appearance and may be impossible to be visualized by the modern medical imaging techniques. On the other hand, the manifestation of TBM in past human populations may differ from that of modern medical cases due in part to the introduction of antibiotics in the treatment of TB; and therefore, probable TB-related bony changes, including GIs, may not occur in recent cases [16,18–19,21]. However, as it has been suggested by Roberts and her colleagues [33], the detailed analysis of well-documented skeletal collections from the pre-antibiotic era can serve as a unique and important basis for determining the diagnostic value of GIs in the paleopathological identification of TBM, since 1) individuals from such collections–who were not treated with antibiotics–can directly be surveyed with macromorphological methods; 2) even subtle bony changes, such as GIs, can be recognized in them; and 3) the manifestation of TBM, and consequently, the appearance of likely TBM-associated lesions may be similar to those of observable in ancient human bone remains.

During the macroscopic evaluation of the 427 selected skeletons with sectioned skulls from the Terry Collection, we found that GIs were ten times more common in individuals recorded to have died of TB than in individuals identified to have died of causes other than TB. Our findings are constituting evidence that there is a positive correlation between GIs and TB. The results of our research project fit in with those of previous studies [e.g., 24–27] concerning the specificity of GIs for TBM, as GIs affected only six individuals in the NTB group (S2 Table). Five out of the above-mentioned six individuals show probable TBM-associated endocranial alterations other than GIs (four cases: abnormally pronounced digital impressions ([e.g., 23–26,71] and one case: periosteal appositions [e.g., 23–26]) (S4 Table) and/or likely TB-related non-endocranial bony changes (two cases: periosteal new bone formations on the visceral surface of ribs [e.g., 16,18–21], two cases: vertebral hypervascularization [e.g., 15,17,32,72–73], two cases: signs of extra-spinal osteomyelitis [e.g., 74–75], and one case: signs of hypertrophic pulmonary osteopathy [e.g., 31,38–40,76]) (S5 Table). It must be noted that even if the recorded cause of death of individuals surveyed in the Terry Collection may not have been TB, individuals could still have suffered from the disease but their death was attributed to another medical condition [18–19]. Moreover, there is always the possibility that an inaccurate cause of death was registered on the morgue record and/or death certificate of individuals from the Terry Collection. Thus, it is possible that in the aforementioned six cases, the observed endocranial and non-endocranial bony changes resulted from TB. In summary, the findings of our study confirm those of Schultz [e.g., 24–26] and Schultz & Schmidt-Schultz [27] that GIs can be considered as pathognomonic features of TBM; and therefore, the paleopathological diagnosis of TBM can be established with a high certainty when GIs are present in ancient human bone remains. The localization pattern and distribution of GIs on the endocranial surface resemble that of the tubercles observed in the affected meninges during the pathogenesis of TBM that further strengthens their tuberculous origin.

Since the beginning of the 21[st] century, a number of molecular evolutionary studies [e.g., 77–82] have improved our knowledge on the origin and evolutionary history of the MTBC, as well as on the co-evolution of its members with the human and various wild and domesticated animal hosts. However, the results of the aforementioned research projects are unfortunately insufficient and controversial [2–4]. Paleomicrobiological analyses of biological remains (e.g., DNA, lipid biomarkers, and ECM proteins) of TB bacteria extracted from skeletons and mummies of people who lived in the past [e.g., 42,83–87] have provided invaluable novel data not only on the evolution of TB but also on its paleoepidemiology throughout prehistoric and historic times. Findings of recent paleoepidemiological studies on human osteoarchaeological series from the pre-antibiotic era [e.g., 88–90] have confirmed the complementarity of paleomicrobiological, microscopic, and traditional, macromorphology-based paleopathological analyses. Their combined application may contribute to facilitating the establishment of a more reliable and accurate paleopathological diagnosis of TB in ancient human bone remains and the assessment of a more relevant frequency of the disease in past human populations [14,91–92].

The above-mentioned examinations require specific scientific knowledge on the macromorphological diagnostics of TB that underlines the importance of our research project, since its results strengthen the tuberculous origin of GIs and imply that they can be considered as specific signs of TBM. Therefore, GIs can be used as diagnostic criteria for TBM in the paleopathological practice, and the diagnosis of TB can be established with a high certainty when GIs are present in ancient human bone remains. Thus, our findings provide paleopathologists with a stronger basis for identifying TB and with a more sensitive means of assessing the frequency of the disease in human osteoarchaeological material. Refinement of macromorphological diagnostic criteria and their application in the paleopathological practice may open new perspectives also in the evolutionary research of TB.

It should be mentioned that because of the composition of the Terry Collection [37], there were no children that could be examined (the youngest individual of the Terry Collection died at the age of 16 years); however, according to the modern medical literature, children under the age of five years represent the most vulnerable group affected by TBM [45,52–53,58–59]. Therefore, the major limitation of our research project was the absence of children in the examined skeletal material. In the future, further investigations on human skeletons of known cause of death from documented collections other than the Terry Collection are necessary to confirm the trends noted in our study. It would also be very useful to examine not only adult but child skeletons of known cause of death to determine whether or not the frequency of GIs is similar to that of observed in adults in the Terry Collection.

Finally, our findings may draw physicians' attention to the rather high prevalence of meningeal involvement in TB patients. This may contribute to further improving the modern medical practice regarding the identification of TBM. Although TBM occurs in less than 1% of all cases with active TB [58] and the vast majority of the individuals in our TB group were identified to have died of pulmonary TB (only one of them was recorded to have died of TBM) (S1 Table), about one-third of them revealed GIs suggestive of TBM on the endocranial surface. Our results fit in with those of autopsy studies revealing that a large number of individuals died of pulmonary TB without developing neurological signs and symptoms exhibited tubercles in the CNS. This indicates that involvement of the CNS in pulmonary TB is quite common [93]. Some recent studies showed that about three-fourths of the patients with CNS TB had pulmonary TB 6–12 months prior to the onset of neurological symptoms [61]. The above-mentioned findings may incite physicians to check pulmonary TB patients for involvement of the CNS even if they do not present with neurological signs and symptoms suggestive of the disease. In such cases, this may facilitate the establishment of an early, accurate diagnosis and

the initiation of a prompt, adequate treatment that are crucial in determining the clinical outcome of TBM.

## Supporting information

**S1 Table. Basic biographic data of individuals in the TB group (N = 234).** (MR = morgue record; DC1 = death certificate primary; DC2 = death certificate secondary; DC3 = death certificate tertiary; c. = circa; F = female; M = male; + = exhibiting GIs;– = not exhibiting GIs). (PDF)

**S2 Table. Basic biographic data of individuals in the NTB group (N = 193).** (MR = morgue record; DC1 = death certificate primary; DC2 = death certificate secondary; DC3 = death certificate tertiary; c. = circa; F = female; M = male; + = exhibiting GIs;– = not exhibiting GIs). (PDF)

**S3 Table.  Distribution of individuals exhibiting GIs in the Terry Collection by affected cranial bones (considering the left and right greater wings of the sphenoid bone as two separate bones), extent, and number of lesions (L = left, R = right).** Number of individuals in the A) TB group and B) NTB group. (PDF)

**S4 Table. Individual data of cases exhibiting GIs regarding other probable TBM-associated endocranial bony changes in the NTB group (Σ = 6).** (APDIs = abnormally pronounced digital impressions; ABVIs = abnormal blood vessel impressions; PAs = periosteal appositions; + = present; − = not present). (PDF)

**S5 Table. Individual data of cases exhibiting GIs regarding possible TB-related non-endocranial bony changes in the NTB group (Σ = 6).** (PNBFs = periosteal new bone formations; HPO = hypertrophic pulmonary osteopathy; + = present; − = not present) (PDF)

**S1 Fig. Demographic profile of individuals surveyed in the Terry Collection.** Number of individuals in the A) TB group (Σ = 234) and B) NTB group (Σ = 193) by age at death and sex. (PDF)

## Acknowledgments

Special thanks go to Luca Kis for the drawing included as Fig 1.

## Author Contributions

**Conceptualization:** Olga Spekker.

**Data curation:** Olga Spekker, David R. Hunt.

**Formal analysis:** Olga Spekker, László Paja.

**Funding acquisition:** Olga Spekker, György Pálfi.

**Investigation:** Olga Spekker.

**Methodology:** Olga Spekker.

**Project administration:** Olga Spekker.

**Resources:** David R. Hunt.

**Supervision:** David R. Hunt, Erika Molnár, György Pálfi, Michael Schultz.

**Visualization:** Olga Spekker, László Paja.

**Writing – original draft:** Olga Spekker, David R. Hunt, Michael Schultz.

**Writing – review & editing:** Olga Spekker, Erika Molnár.

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
