## [Decision Letter · Decision Letter 0]

10 Jan 2020

PONE-D-19-27108

Tracking down the White Plague: The skeletal evidence of tuberculous meningitis in the Robert J. Terry Anatomical Skeletal Collection

PLOS ONE

Dear Dr. Spekker,

Thank you for submitting your manuscript to PLOS ONE. After careful consideration, we feel that it has merit but does not fully meet PLOS ONE’s publication criteria as it currently stands. Therefore, we invite you to submit a revised version of the manuscript that addresses the points raised during the review process.

As suggested by the reviewers, please focus on re-structuring the manuscript by removing redundant information and checking thoroughly for English language. Address all the reviewer comments on the manuscript in addition to the rebuttal letter.

We would appreciate receiving your revised manuscript by Feb 24 2020 11:59PM. To enhance the reproducibility of your results, we recommend that if applicable you deposit your laboratory protocols in protocols.io, where a protocol can be assigned its own identifier (DOI) such that it can be cited independently in the future. For instructions see: http://journals.plos.org/plosone/s/submission-guidelines#loc-laboratory-protocols

We look forward to receiving your revised manuscript.

Kind regards,

Selvakumar Subbian, Ph.D.

Academic Editor

PLOS ONE

Journal Requirements:

2. In your manuscript, please provide additional information regarding the specimens used in your study. Ensure that you have reported specimen numbers and complete repository information, including museum name and geographic location.

For more information on PLOS ONE's requirements for paleontology and archaeology research, see " ext-link-type="uri" xlink:type="simple">https://journals.plos.org/plosone/s/submission-guidelines#loc-paleontology-and-archaeology-research."

Reviewers' comments:

Reviewer's Responses to Questions

**Comments to the Author**

1. Is the manuscript technically sound, and do the data support the conclusions?

Reviewer #1: Yes

Reviewer #2: Yes

2. Has the statistical analysis been performed appropriately and rigorously? 

Reviewer #1: Yes

Reviewer #2: Yes

3. Have the authors made all data underlying the findings in their manuscript fully available?

Reviewer #1: Yes

Reviewer #2: Yes

4. Is the manuscript presented in an intelligible fashion and written in standard English?

Reviewer #1: Yes

Reviewer #2: Yes

5. Review Comments to the Author

Reviewer #1: Nicely written though extensive manuscript. All the sections are clear and focus on significant points of discussion. It seems repetitions of the text. Authors can significantly reduce words count by deleting the repetitions.

Reviewer #2: This is an interesting study of skeletal changes related to tuberculosis, using a well-characterized collection. The authors identify that a particular finding- granular impressions in the skull- was present in 29% of skeletons where TB was the cause of death, compared with 3% without TB disease. There are numerous strengths to this study, particularly the blinding of study personnel in reviewing these skulls, and the detailed methodology provided. The results are surprising, but support the authors' contention that CNS TB disease is more common than reported in clinical studies of TB presentation.

Major comments:

-I would take issue with the association of these GIs with a diagnosis of "TBM", more accurate would be to refer to these findings as evidence of CNS TB. The formation of CNS tuberculomas precedes, in all instances, with the pathophysiologic events of TBM- namely, tuberculoma rupture and inflammatory response in the meninges. While some of these patients may have had referable signs/symptoms of TBM, many may have had clinically silent CNS tuberculomas. That is an important distinction, as we do not routinely perform the types of diagnostic studies that would detect these lesions among pulmonary TB patients. The absence of TBM relapses post pulmonary TB treatment suggests that these lesions are adequately treated with the standard TB drug regimen. I recommend making the distinction throughout the manuscript that you are identifying/evaluating a marker of CNS TB.

-If possible, it would be interesting to examine for the types of changes that would characterize the transition from CNS TB to TBM, including some of the features associated with CNS vascular disease that are described. The subset of CNS TB patients who have developed TBM could be identified with these additional features, perhaps that is a separate study, but this concept is presented briefly in the manuscript and could be developed in greater detail, or at least clarified.

-The discussion should include a single paragraph that summarizes all the limitations, which are scattered a bit. A chief limitation would be the absence of children, as we know that CNS TB is more common among young children. It would be very useful to validate these findings in this age cohort, where we would expect the proportion of skulls with GIs to be even greater than observed here among adults.

Minor comments

-Line 73- often, there are no identifiable risk factors for the re-activation of TB disease from latency, this process remains poorly understood.

Line 80- all patients with latent TB are at risk for progressing to active TB, but 5-15% of patients (without LTBI treatment) do progress.

-Line 85-90- not sure of the relevance of this discussion of drug resistant TB, given the focus on the pre-antibiotic era.

-Line 138- is this true? wouldn't the CNS tuberculomas, slowly enlarging over time, be the incipient factor, the cause rather than the effect of TBM.

-Line 216- would emphasize here that the skeletal reviewer was "blinded" as to the cause of death.

-The discussion is quite length, includes much information tangential to the project (e.g. Line 340 onwards), or re-states findings from the results (Line 397 onwards). The manuscript would be improved with substantial editing of these sections.

-In my view, the take home message gets lost and should be more emphasized- CNS TB disease, as detected by the GIs in skull from the pre-antibiotic era, is more common than previously recognized.

6. PLOS authors have the option to publish the peer review history of their article (what does this mean?). If published, this will include your full peer review and any attached files.

Reviewer #1: No

Reviewer #2: Yes: Christopher Vinnard

---

## [Author Response · Author response to Decision Letter 0]

25 Feb 2020

Dear Dr. Selvakumar Subbian,

I am very thankful for the reviewers’ insightful and constructive comments regarding our manuscript entitled “Tracking down the White Plague: The skeletal evidence of tuberculous meningitis in the Robert J. Terry Anatomical Skeletal Collection” that was submitted to PLOS ONE (manuscript ID: PONE-D-19-27108). I am sure that they helped us to improve the quality of our manuscript. The main text has been modified following the reviewers’ suggestions, and the revised version of our manuscript has been uploaded to the submission site of PLOS ONE.

Responses to major comments:

1) Reviewer 2 noted that it would be more accurate to refer to GIs as evidence of CNS TB instead of evidence of TBM. It should be mentioned that unfortunately, the paleopathological diagnosis of a disease has a number of limitations. One of the major limitations is that in most paleopathological studies only bone remains are available for examination; and therefore, only those diseases can be diagnosed that directly or indirectly affect the bones. CNS TB has many forms (e.g., TBM, tuberculomas, and TB abscesses) but not all of them can leave traces on the cranial bones that could be detected. We agree with Reviewer 2 that the formation of meningeal, subpial or subependymal tuberculous granulomas (also known as tubercles or Rich foci) precedes with the pathophysiologic events of TBM and that TBM usually develops subsequent to the rupture of one or more tubercles. In later stages of TBM, when tubercles form not only in the leptomeninges but also in the outermost meningeal layer, there is a chance that the disease can be diagnosed by macromorphological examination of the cranial bones, because the tubercles formed in the outermost meningeal layer are in direct contact with the inner surface of the skull. The pressure exerted by these tubercles on the endocranial surface can result in the development of GIs that can be detected even with the naked eye. However, in earlier stages of TBM, when the outermost meningeal layer is not affected, or in case of TB mass lesions (i.e., tuberculomas and TB abscesses) that are located deep in the brain parenchyma and result from enlargement and/or fusion of one or more tubercles, CNS TB cannot be diagnosed by macromorphological examination of the cranial bones, since these lesions have no direct contact with the inner surface of the skull. Therefore, they cannot lead to the development of bony changes on the endocranial surface. It should also be mentioned that even if there is an epidural tuberculoma or TB abscess that are in direct contact with the inner surface of the skull, the bony changes that could result from them would differ from GIs. Although similar to the tubercles, an epidural tuberculoma could lead to the development of an impression on the endocranial surface, its size would be bigger than that of a GI (its size and shape would correspond not to a single tubercle but to the tuberculoma that resulted in its development). An epidural TB abscess would more likely lead to an erosive bone lesion (even with perforation of the skull). Consequently, based on the presence of GIs on the inner surface of the skull, we can diagnose only TBM and only in its later stages, when the pathological process affects not only the leptomeninges but also the outermost meningeal layer. It must also be noted that the absence of GIs does not mean that an individual did not have TBM. On the one hand, patients can die before tubercles appear in the outermost meningeal layer, and consequently before the development of GIs. On the other hand, even if patients survive until tubercles form in the outermost meningeal layer, at least a couple of weeks after the formation of these tubercles is required for the development of GIs on the endocranial surface and it is not sure that patients survive until that. Therefore, even if we consider GIs as diagnostic criteria for TBM in the paleopathological practice, we will still very likely underestimate the frequency of the disease in the examined skeletal material, because we can diagnose only those cases, in which GIs are present. Based on the above, if Reviewer 2 agrees with us, we would not like to refer to GIs as diagnostic criteria for CNS TB but as diagnostic criteria only for TBM. 

2) Reviewer 2 commented that if possible, it would be interesting to examine for the types of changes that would characterize the transition from CNS TB to TBM, including some of the features associated with CNS vascular disease. We agree with Reviewer 2 that with additional features, we could further improve the diagnosis of TBM in human osteoarchaeological series. In the paleopathological literature, not only GIs but other endocranial changes – i.e., abnormally pronounced digital impressions (APDIs), abnormal blood vessel impressions (ABVIs), and periosteal appositions (PAs) – were also mentioned in relation to TBM. During our examinations in the Terry Collection, the macromorphological characteristics and the frequency of APDIs, ABVIs, and PAs were also evaluated, these findings can be found in detail in the first author’s PhD dissertation (http://doktori.bibl.u-szeged.hu/9714/1/PhD_dissertation_SO.pdf). We would like to publish our results on APDIs, ABVIs, and PAs in separate articles; therefore, we did not include these findings and did not want to develop these aspects in greater detail in our PLOS ONE manuscript. (The manuscript about APDIs is already under review, whereas the manuscript about ABVIs and PAs will be submitted soon.)

3) Reviewer 2 noted that the Discussion and Conclusions part should include a paragraph that summarizes all the limitations, which are scattered a bit in the original manuscript. Reviewer 2 also mentioned that a chief limitation would be the absence of children, as we know that CNS TB is more common among young children. Following Reviewer 2’s instructions, a paragraph about the major limitation of our study (i.e., the absence of children in the examined skeletal material) has been included in the Discussion and Conclusions part. We agree with Reviewer 2 that it would be very useful to validate our findings in children. Unfortunately, because of the composition of the Terry Collection, there were no children that could be examined, because the youngest individual of the Terry Collection died at the age of 16 years. Nonetheless, we hope that in the future, we will have the opportunity to evaluate not only adult but child skeletons of known cause of death in other documented collections and thereby determine whether or not the frequency of GIs is similar to that of observed in adults in the Terry Collection. We must agree with Reviewer 2 that from the clinical point of view, besides the absence of children, our study must have other limitations, as well. Nevertheless, it should be noted that our study is primarily a paleopathological study. From the paleopathological point of view, studies in which the skeletal material comes from human osteoarchaeological series have a number of limitations (e.g., unknown age at death, sex, cause of death, and “race”; missing bones; fragmentary bone remains; damaged bone surfaces). By choosing a documented skeletal collection as the skeletal material for our study, we attempted to avoid the aforementioned limitations.

Responses to minor comments:

1) Regarding line 73, Reviewer 2 mentioned that often, there are no identifiable risk factors for the reactivation of TB disease from latency, and this process remains poorly understood. Following Reviewer 2’s comment, this section of the Introduction part has been corrected and supplemented. 

2) Concerning line 80, Reviewer 2 noted that all patients with latent TB are at risk for progressing to active TB, but 5–15% of patients (without LTBI treatment) progress. We must agree with Reviewer 2 that we did not appropriately phrased our sentence in the original manuscript. Following Reviewer 2’s comment, the sentence has been corrected.

3) As for lines 85–90, Reviewer 2 was not sure of the relevance of the discussion of drug resistant TB. We agree with this comment and the irrelevant information has been deleted from this paragraph.

4) Regarding line 138, Reviewer 2 asked if wouldn’t the CNS tuberculomas, slowly enlarging over time, be the incipient factor, the cause rather than the effect of TBM. We agree with Reviewer 2 that TBM usually results from the rupture of meningeal, subpial or subependymal tubercles. In our manuscript, we did not want to state that tubercles are not the cause but the effect of TBM. What we wanted to say is that GIs can be established by pressure atrophy of tubercles. Unfortunately, in paleopathological studies, in which only bone remains are available, tubercles themselves cannot be detected. Nevertheless, during later stages of TBM, when tubercles can develop not only in the leptomeninges but also in the outermost meningeal layer, there is a chance that these tubercles leave their traces on the endocranial surface. Since the outermost meningeal layer is directly adherent to the inner surface of the skull, the tubercles formed in it are also in direct contact with the endocranial surface. These tubercles exert pressure on the inner surface of the skull that can lead to bone atrophy, i.e., to the development of GIs on the endocranial surface.

5) Concerning line 216, Reviewer 2 mentioned that it should be emphasized that the skeletal reviewer was “blinded” as to the cause of death. Following Reviewer 2’s suggestion, this section of the Methods part has been rephrased to emphasize that the study personnel had no information on the cause of death of the examined individuals during the macromorphological analysis of the selected skulls.

6) Reviewer 1 noted that the original manuscript contains repetitions that should be deleted from the text. Reviewer 2 also mentioned that the Discussion and Conclusions part of the original manuscript is quite long, and it includes much information tangential to the project, or re-states findings from the Results part. We agree with the reviewers’ comments and following their instructions, sentences containing irrelevant information or repetitions have been deleted from the main text.

7) Reviewer 2 noted that the take home message gets lost and should be more emphasized. We must agree with Reviewer 2 that we did not appropriately highlighted the take home message of our study in the original manuscript. Following Reviewer 2’s comment, in the revised version of our manuscript, the take home message has been more emphasized in the Discussion and Conclusions part. Nonetheless, it must be noted that in our opinion, the take home message of our study is not the one that Reviewer 2 mentioned in his review (i.e., CNS TB disease, as detected by the GIs in skull from the pre-antibiotic era, is more common than previously recognized). Our study is primarily a paleopathological study, and its main aim was to contribute to strengthening the diagnostic value of GIs in the identification of TB in human osteoarchaeological series. (Although GIs were used as diagnostic criteria for TBM in the paleopathological practice since the late 20th century, their diagnostic value has been questioned.) Therefore, from the paleopathological point of view, the take home message of our study is that GIs can be considered as specific signs of TBM, and they can be used as diagnostic criteria for the disease in the paleopathological practice. Of course, we agree with Reviewer 2 that from the clinical point of view, the most important and relevant finding of our study is that the tuberculous involvement of the CNS is more common than previously thought. We were a little surprised but – at the same time – very happy when we realized that our results can have relevance not only to the paleopathological practice but also to the modern medical practice. Nevertheless, for now, we could examine only the Terry Collection, and in the future, further investigations on other documented skeletal collections are necessary to confirm the trends noted in our study.

In the revised version of our manuscript, we tried to execute all suggestions of the reviewers. I hope this new version will be suitable for publication in PLOS ONE. 

Thank you again for the reviewers’ insightful and constructive comments and your editorial work!

Sincerely yours,

Dr. Olga Spekker, PhD

---

## [Editor Report · Decision Letter 1]

2 Mar 2020

Tracking down the White Plague: The skeletal evidence of tuberculous meningitis in the Robert J. Terry Anatomical Skeletal Collection

PONE-D-19-27108R1

Dear Dr. Spekker,

We are pleased to inform you that your manuscript has been judged scientifically suitable for publication and will be formally accepted for publication once it complies with all outstanding technical requirements.

With kind regards,

Selvakumar Subbian, Ph.D.

Academic Editor

PLOS ONE
---

## [Editor Report · Acceptance letter]

5 Mar 2020

PONE-D-19-27108R1 

Tracking down the White Plague: The skeletal evidence of tuberculous meningitis in the Robert J. Terry Anatomical Skeletal Collection 

Dear Dr. Spekker:

I am pleased to inform you that your manuscript has been deemed suitable for publication in PLOS ONE. Congratulations! Your manuscript is now with our production department. 

With kind regards,

on behalf of

Dr. Selvakumar Subbian 

Academic Editor

PLOS ONE